# Cone Dystrophy Associated with a Novel Variant in the Terminal Codon of the *RPGR*-*ORF15*

**DOI:** 10.3390/genes12040499

**Published:** 2021-03-29

**Authors:** Vlasta Hadalin, Maja Šuštar, Marija Volk, Aleš Maver, Jana Sajovic, Martina Jarc-Vidmar, Borut Peterlin, Marko Hawlina, Ana Fakin

**Affiliations:** 1Eye Hospital, University Medical Centre Ljubljana, Grablovičeva 46, 1000 Ljubljana, Slovenia; vlasta.hadalin@gmail.com (V.H.); sustar.majchi@gmail.com (M.Š.); jana.sajovic@gmail.com (J.S.); martina.jarcvidmar@gmail.com (M.J.-V.); marko.hawlina@gmail.com (M.H.); 2Clinical Institute of Medical Genetics, University Medical Centre Ljubljana, Šlajmerjeva ulica 4, 1000 Ljubljana, Slovenia; marija.volk@kclj.si (M.V.); ales.maver@kclj.si (A.M.); borut.peterlin@guest.arnes.si (B.P.)

**Keywords:** RPGR, ORF15, cone-dystrophy

## Abstract

Mutations in *RPGR^ORF15^* are associated with rod-cone or cone/cone-rod dystrophy, the latter associated with mutations at the distal end. We describe the phenotype associated with a novel variant in the terminal codon of the *RPGR^ORF15^* c.3457T>A (Ter1153Lysext*38), which results in a C-terminal extension. Three male patients from two families were recruited, aged 31, 35, and 38 years. Genetic testing was performed by whole exome sequencing. Filtered variants were analysed according to the population frequency, ClinVar database, the variant’s putative impact, and predicted pathogenicity; and were classified according to the ACMG guidelines. Examination included visual acuity (Snellen), colour vision (Ishihara), visual field, fundus autofluorescence (FAF), optical coherence tomography (OCT), and electrophysiology. All patients were myopic, and had central scotoma and reduced colour vision. Visual acuities on better eyes were counting fingers, 0.3 and 0.05. Electrophysiology showed severely reduced cone-specific responses and macular dysfunction, while the rod-specific response was normal. FAF showed hyperautofluorescent ring centred at the fovea encompassing an area of photoreceptor loss approximately two optic discs in diameter (3462–6342 μm). Follow up after 2–11 years showed enlargement of the diameter (avg. 100 μm/year). The novel c.3457T>A (Ter1153Lysext*38) mutation in the terminal *RPGR^ORF15^* codon is associated with cone dystrophy, which corresponds to the previously described phenotypes associated with mutations in the distal end of the *RPGR^ORF15^*. Minimal progression during follow-up years suggests a relatively stable disease after the initial loss of the central cones.

## 1. Introduction

### 1.1. Molecular Genetics of RPGR

*RPGR* (retinitis pigmentosa GTPase regulator) is a gene located on the X chromosome with several different protein coding transcripts, expressed in various tissues. The two major *RPGR* isoforms are a constitutive *RPGR^ex1–19^*, derived from exons 1–19, encoding a protein of 815 amino acids; and *RPGR^ORF15^*, which shares exons 1–14 with the constitutive isoform and contains a large *ORF15* as its terminal exon, encoding a protein of 1152 amino acids [1] (illustrated in Figure 1). Both are expressed in the retina, with the *RPGR^ORF15^* being the predominant one [2].

### 1.2. Function of the RPGR Protein

The function of the RPGR protein has not yet been fully elucidated. The RPGR^ex1–19^ isoform is widely expressed in tissues [2,3,4] and is found within cells at the transition zone of primary and motile cilia [5,6], or at centrosomes and their constituent centrioles in dividing cells [7]. In the retina, the RPGR^ex1–19^ isoform localises to the developing and mature photoreceptor connecting cilium (CC), and shows a slightly different developmental expression pattern and affinity for the axonemal fraction compared with the other major RPGR isoform RPGR^ORF15^, suggesting overlapping, but also distinct, functions [8]. The RPGR^ex1–19^ transcript encodes a predicted 90kDa protein. The C-terminus of the RPGR^ex1–19^ has an isoprenylation signal, suggesting that this isoform is membrane bound, consistent with its reported attachment to endoplasmic reticulum membranes in addition to its presence at the connecting cilium [9]. There has also been suggested a localization of RPGR^ex1–19^ isoform to the Golgi [10] and other compartments in the inner segment that may indicate a possible involvement of RPGR in post-Golgi sorting of cargo-containing vesicles, phototransduction components, and other outer segment proteins towards the connecting cilium [7,9,11]. The RPGR^ORF15^ isoform is predicted to be 127 kDa and exon *ORF15* includes an acidic, repetitive, glutamic acid/glycine-rich domain and a basic C-terminal domain [12]. The repetitive domain length is varying considerably among species [2,13]. In contrast, the basic C-terminal domain is highly conserved among vertebrates [14]. In dividing cells, RPGR^ORF15^ is present in centrosomes, while in nondividing cells containing primary cilia it is found in the transition zone of the ciliary axoneme, as mentioned, the equivalent structure to the photoreceptor connecting cilium [5,14,15]. Some studies suggest that RPGR^ORF15^ binds to the basal body and the axoneme [7,16]. Other studies suggest that RPGR^ORF15^ is present in the photoreceptor outer segment (OS), however the studies are inconclusive, possibly due to differences between laboratories because of the use of different antibodies, tissue processing procedures, or species differences in the OS structure [17,18]. The exons 1–14 of both isoforms encode a structure similar to regulator of chromosome condensation 1 (RCC1) at the N-terminus [3,19]. RCC1 is a well characterised protein that functions as a guanine nucleotide exchange factor for Ran (a Ras-related nuclear protein) and is thought to play an important role in nucleocytoplasmic transport and regulation of cell division as well as the predominant ciliary localization [1,7,20]. RanGTP serves as an energy source for molecular motors that move cargo through the nuclear pore complex [11]. The resultant high concentration of RanGTP in the connecting cilium as generated by RPGR could enable a putative RanGTP-dependent process that drives unidirectional movement of opsins and other cargo across the connecting cilium to the outer segment [21,22]. RPGR-interacting protein (RPGRIP1) localises to the connecting cilia and is thought to hold RPGR in this location because it consists of an N-terminal region predicted to form coiled coil structure linked to a C-terminal tail that binds RPGR [23]. RPGR interacts with several other ciliary proteins, some of which have also been associated with cone/cone-rod dystrophy (reviewed in Discussion).

### 1.3. Phenotypes Associated with Mutations in RPGR

To date, over 300 disease-causing variants have been identified in *RPGR* [2,24]. All were found in the shared exons of both isoforms and the exon *ORF15*, while none were reported in the exons 16–19 [25]. The *ORF15* exon contains an unusual repetitive sequence encoding 567 amino acids rich in glycine and glutamic acid residues that is considered to be a ‘mutational hot spot’ [2].

Two distinctly different phenotypes have been recognised in *RPGR* patients, the more frequent primarily affecting rods (retinitis pigmentosa, RP), and the other primarily affecting cones (cone/cone-rod dystrophy) [26,27]. Myopia is a common feature in both [27,28]. *RPGR*-associated RP is one of the most frequent (over 70% of X-linked RP [26]) and most severe forms of retinitis pigmentosa [29]. It is characterized with presentation in childhood, with first reported symptoms being nyctalopia and peripheral visual loss [27]. *RPGR* is the causative gene in approximately 1% cases of cone/cone-rod dystrophy [30,31]. It is characterised by central visual loss (reduced acuity, colour vision, and central scotoma), and in some patients photophobia [29]. The patients with rod system dysfunction also report night blindness and may exhibit peripheral visual field loss [29]. Most patients have myopia, with 50–72% having a refractive error of greater than −6 dioptres [27,28]. ERG in cone dystrophy typically shows delayed and reduced light adapted (LA) ERGs and, in cone-rod dystrophy also abnormal dark adapted (DA) ERGs [32]. There is typically early and severe macular involvement in cone/cone-rod dystrophy cases, characterised by pattern ERG reduction, although pattern ERGs might be relatively high in young or mild cases, with relatively small rings of increased parafoveolar FAF [32]. Due to the X-linked recessive inheritance of *RPGR*, the disease predominantly affects males, however, a retinal phenotype can also be seen in female carriers, caused by the inactivation of the normal X chromosome, the severity depending on the degree of normal X chromosome inactivation [33].

Fundus autofluorescence imaging (FAF) in *RPGR* retinopathy often reveals parafoveal rings of increased autofluorescence, delineating the border between the affected and unaffected retina [27,30,32,34,35] and can be used to follow the disease progression. In RP patients, the area within the ring corresponds to the preserved part of the retina [34,36]. This is common to RP of different genetic background and many studies have shown the progressive constriction of the ring with time [37,38,39,40]. On the contrary, the rings in cone/cone-rod dystrophy encompass the area of degenerating retina [27,30,32]. Studies have shown increasing of the ring diameter with time in *RPGR* cone dystrophy cases [31,35,41].

It is not known why the *RPGR* mutations result in two contrasting disorders. It seems that the phenotype depends on the location of the mutation: Mutations in the exons 1–14 and the proximal part of the *ORF15* exon usually result in retinitis pigmentosa, while the mutations in the distal end of the *ORF15* exon cause cone/cone-rod dystrophy [11,29,42,43,44]. There is however a watershed zone of approximately 100 aminoacids between the two regions, where mutations can result in either phenotype, even within the same family [29], the reasons for which are not understood. It is also not clear whether the cone and cone-rod subtypes also depend on the mutation location.

We present the clinical characteristics of three patients harbouring a novel C-terminal extension variant resulting in cone dystrophy due to the loss of the terminal codon of the *RPGR^ORF15^*, c.3457T>A (Ter1153Lysext*38).

## 2. Materials and Methods

### 2.1. Patients

The study included three male patients from two families aged 31, 35, and 38 years, ascertained from the Eye Hospital University Medical Centre Ljubljana, Slovenia. The study was conducted in agreement with the Declaration of Helsinki. Informed written consent was obtained from the patients.

### 2.2. Genetic and Bioinformatic Analysis

Genetic analysis was performed in a proband from each family (Patient 1 and Patient 2). Genomic DNA was extracted from blood samples according to the standard procedure. Whole exome sequencing was performed. Sequencing of the defined clinical target was performed using next-generation sequencing on the isolated DNA sample. Briefly, the fragmentation and enrichment of the isolated DNA sample were performed according to the Illumina Nextera Coding Exome capture protocol, with subsequent sequencing on Illumina NextSeq 550 in 2 × 100 cycles (Illumina, San Diego, CA, USA). After duplicates were removed, the reads were aligned to the UCSC hg19 reference assembly using the BWA algorithm (v0.6.3) and variants were called using the GATK framework (v2.8). Only variants exceeding the quality score of 30.0 and depth of 5 were used for down-stream analyses. Variant annotation was performed using ANNOVAR and snpEff algorithms, with pathogenicity predictions in dbNSFPv2 database. Reference gene models and transcript sequences are based on the RefSeq database. Structural variants were assessed using CONIFER v0.2.2 algorithm. Variants with population frequency exceeding 1% in gnomAD, synonymous variants, intronic variants and variants outside the clinical target were filtered out during analyses. An in-house pipeline was used for bioinformatic analyses of exome sequencing data, in accordance with GATK best practice recommendations [45]. The interpretation of sequence variants was based on ACMG/AMP standards and guidelines [46]. Sequencing the DNA sample, we reached median coverage of 67× and covered over 99.9% targeted regions with minimum 10× depth of coverage [47]. The presence of the mutation in the population was examined in the gnomAD database (gnomad.broadinstitute.org, accessed on 06.01.2021). Single-nucleotide polymorphism (SNP) analysis comparing X-chromosomes of probands from each family was performed in order to inspect the possibility of a founder effect.

### 2.3. Clinical Examination

An accurate family history was recorded, and all patients underwent a complete ophthalmic examination, which included best-corrected visual acuity (Snellen), slit lamp biomicroscopy, and dilated fundus examination. Retinal fundus photographs were obtained by conventional 35° fundus colour photographs (Topcon, Tokyo, Japan). Fundus autofluorescence imaging (FAF) (30° and 55° of the central retina) and optical coherence tomography (OCT) extending 8 mm of the macula was performed with a confocal scanning laser ophthalmoscope (Spectralis; Heidelberg Engineering, Heidelberg, Germany). The horizontal diameters and areas of the hyperautofluorescent rings on FAF were measured using an automated viewing module available in the Spectralis software. The outer border of the ring was used for the measurement. The integrity of the photoreceptors was determined by qualitatively assessing the inner segment ellipsoid (ISe) band of the photoreceptors on the OCT. Full-field electroretinography (ffERG) and pattern electroretinography (PERG) were performed to incorporate the International Society for Clinical Electrophysiology of Vision Standards [48,49].

## 3. Results

### 3.1. Genetic Findings

Genetic testing in the probands from two independent Slovenian families identified a novel variant c.3457T>A (Ter1153Lysext*38) in the *RPGR^ORF15^*. The variant is predicted to disrupt the terminal codon of the *RPGR^ORF15^,* resulting in a C-terminal extension of the protein by 38 aminoacids ending with a new stop codon. The variant has not been reported in the biomedical literature and is not present in the GnomAD population database (gnomad.broadinstitute.org, accessed on 06.01.2021). Single-nucleotide polymorphism analysis comparing X-chromosomes of the probands is shown in Figure 2C. On the X-chromosome we identified a large block of hemizygous SNPs shared by both probands with an estimated size of 29 megabases, which suggests that observation of an identical variant in two independent families may be due to a founder effect.

### 3.2. Clinical Presentation

The family pedigrees and clinical characteristics of the 3 male patients are shown in Table 1 and Figure 2, respectively. All had myopia from childhood and adult-onset cone dystrophy. The visual loss appeared in the early 30s in all three, consisting of loss of the central vision (N = 3), photophobia (N = 2), and/or loss of colour discrimination (N = 2). One patient (Patient 2) also reported night vision difficulties. Patient 1 additionally suffered from recurring rhegmatogenous retinal detachment on the left eye at 16 and 31 years of age. At the first exam at the clinic for retinal dystrophies at the median age of 35 years (range, 31–38 years) the median best-corrected visual acuity on the better eye was 1.5 logMAR (range 0.2–1.8; decimal Snellen 0.03; range 0.015–0.6). All had central scotoma while fundus examination revealed mottled pigment in the macula, arteriolar attenuation and optic disc pallor. Pigment formation in the macula and periphery was observed in the left eye of Patient 1 as a remnant of the retinal detachment. Fundus autofluorescence in all eyes showed hyperautofluorescent rings centred at the fovea, encompassing an area of reduced autofluorescence approximately two optic discs in diameter (3462–6342 μm). The rings delineated the loss of the outer retina in the fovea on OCT (Figure 1). Electrophysiology showed significantly reduced to undetectable PERG and/or mfERGs, corresponding to the loss of macular function, normal to borderline dark-adapted ERG, and significantly reduced to undetectable light-adapted ERG (Figure 3). The patients were followed for a median time of 3 years (range, 2–8 years). At their latest exam at the median age of 38 years (range 33–49 years), the median BCVA was 1.5 logMAR (range 0.5–1.5; Snellen 0.03; range 0.03–0.3). FAF showed enlargement of the hyperautofluorescent ring diameters of about 100 μm per year (range, 31–194 μm); while the enlargement of the ring area was calculated to be about 0.33 mm^2^ per year (range, 0.15–0.63 mm^2^) (Figure 4).
Figure 1Retinal imaging (colour fundus, FAF, OCT) and *RPGR^ORF15^* scheme. Colour fundus images (1–6, 1′–6′ represent follow-up), corresponding FAF (a–f, a’–f’ represent follow-up) and ring enlargement demonstration (a–f, a’–f’ represent follow-up), corresponding OCT (1–6, 1′-6′ represent follow-up) of all patients included in the study, from their first and last examination. Colour fundus images in BE of all the patients show Bull’s eye appearance of the macula, optic pallor, and attenuated vessels; Patient 1 in his LE (2 and 2′) also presents bone spicules in the central and peripheral retina due to retinal detachment. FAF in BE of all the patients show the hyperautofluorescent ring; Patient 1 in his LE (b and b’) also presents RPE mottling in the inferior temporal retina (after retinal detachment). All the patients showed the hyperautofluorescent ring enlargement displayed in Figure 4. OCT in BE of all the patients shows absent RPE, Ise, and ELM in the central macula; Patient 2 in his LE (d and d’) and Patient 3 in his BE (e and e’, f and f’) present remnants of ELM in the foveola. In the right part, schematic representation of the amino acid structure of the *RPGR^ORF15^* and the predicted effect of the mutation c.3457T>A (Ter1153Lysext*38) elongating the protein by 38 aminoacids, is being presented.
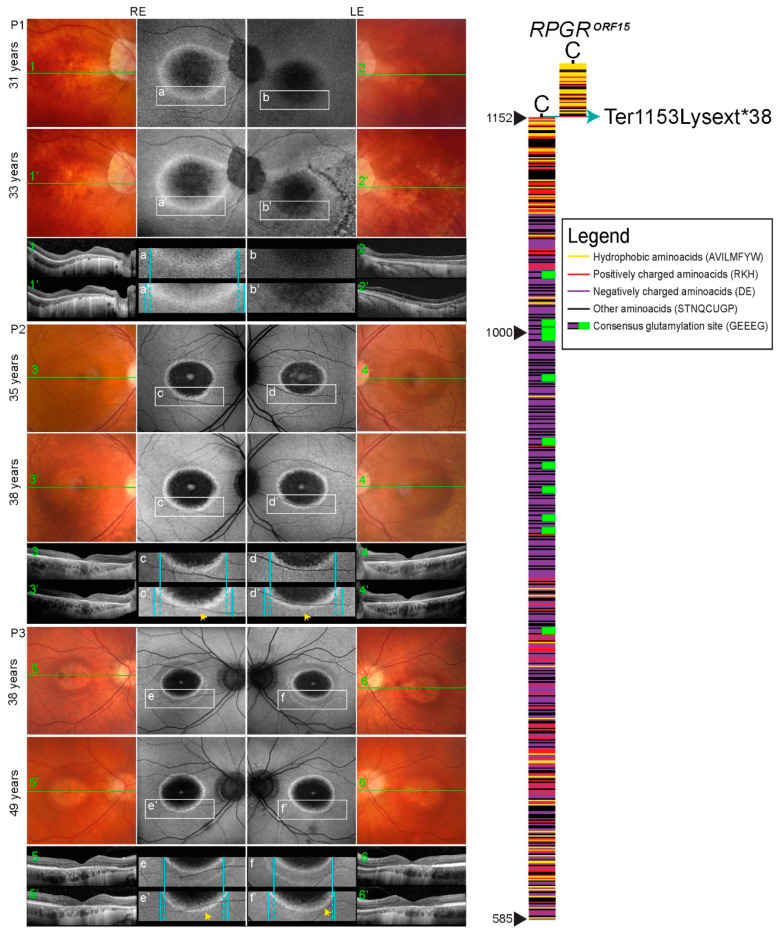

Figure 2Pedigrees of two families harbouring the *RPGR^ORF15^* c.3457T>A mutation and single-nucleotide polymorphism (SNP) analysis. (**A**)—family of Patient 1 (III-1), (**B**)—family of Patients 2 (II-1) and 3 (III-3). Probands from each family are marked with an arrow. Other living relatives were not affected; the mother of Patient 1 had mild myopia. (**C**)—Single-nucleotide polymorphism analysis comparing X-chromosome variants of Patient 1 (Family 1) and Patient 2 (Family 2) has been performed in order to inspect the possibility of a founder effect. The X-chromosomes differed in the majority of the segments, except in one where they involved the same hemizygous variants (enlarged segment at the bottom of the Figure). The findings suggest that the studied variant is being transmitted on the same haplotype and our patients could have inherited the variant from a shared ancestor.
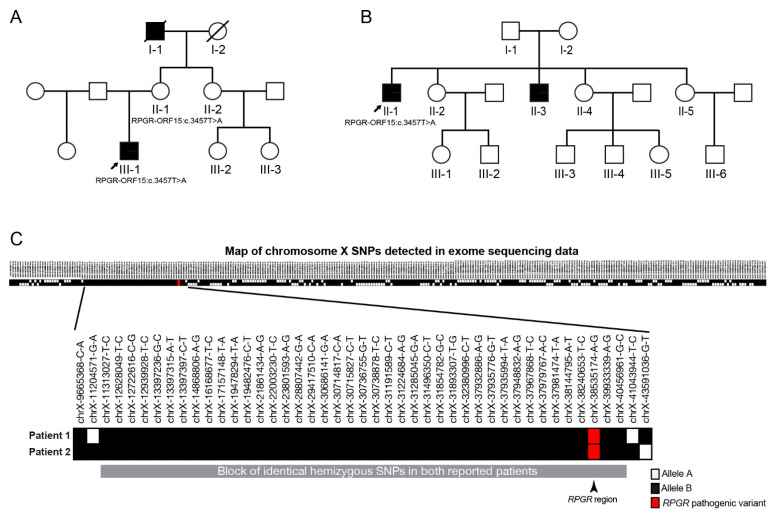



## 4. Discussion

### 4.1. Phenotype Associated with the Terminal Mutation in RPGR^ORF15^

The novel C-terminal extension variant c.3457T>A (Ter1153Lysext*38) due to the loss of the terminal *RPGR^ORF15^* codon was associated with myopia and adult onset cone dystrophy in all three patients.

It has been previously shown that mutations in the proximal part of the *RPGR^ORF15^* result in RP while mutations in the distal end result in cone and cone-rod dystrophy [42,44,50,51,52]. It has not yet been elucidated however whether the latter two phenotypic subtypes occur interchangeably or are also location specific. Considering that the mutation in the last codon resulted in cone-dystrophy in all three patients we hypothesize that the involvement of rods diminishes downstream of the *ORF15* exon. It is possible that the cone involvement increases concomitantly with the decrease of rod involvement or else is present at the same level in all phenotypes and becomes more prominent with diminished rod involvement. Sandberg et al. reported faster deterioration of visual acuity in patients with *RPGR*-associated RP in comparison to RP caused by variants in other genes, at around 4–5% per year (in comparison to *RHO* patients with decline of 1.6% per year), which has been attributed to a greater rate of foveal thinning and outer nuclear layer loss [53]. This may suggest an additional direct cone involvement already in the *RPGR*-RP phenotype. Studies have however shown a faster rate of visual acuity decline in *RPGR*-cone-rod patients in comparison to the *RPGR*-RP patients with approximately half of the cone-rod patients reaching legal blindness at the age of 40–50 [27,28], whereas patients with RP were reported to reach legal blindness at a median age of 77 [53]. This is comparable to our observation, as among our three patients, two were legally blind before the age of 40 (at the ages 31 and 38 years) whereas Patient 2 had still some functional central vision at the age of 38 years.

Electrophysiology can be used to distinguish between cone and cone-rod dystrophy. Light-adapted ERGs are typically abnormal in both, whereas in cone-rod dystrophy, there is an additional abnormality of the dark-adapted ERGs. There is typically early and severe macular involvement in both, which is characterised by pattern ERG and multifocal ERG reduction, where pattern ERG responses might be relatively high in young or mild cases, with relatively small rings of increased parafoveal FAF [32]. All three patients studied had abnormal PERG and/or mfERG corresponding to the photoreceptor loss in the macula. All three also had severely abnormal light-adapted ERGs, indicating severe generalised cone dysfunction. Patient 2, who had the best visual acuity and lowest refraction error, also had better LA ERG and residual PERG response (Figure 3). Their rod-specific responses were normal. However, the dark-adapted bright flash ERGs exhibit borderline amplitudes in 2 patients. Due to a combined contribution of both cone and rod system activity to this response, its borderline abnormality was most probably a consequence of severe cone dysfunction. This was previously shown in stationary cone dysfunction syndromes, namely in complete achromatopsia [54], although it is not yet clear whether a lack of functional cones might additionally affect rods or their neural pathways [55]. Additionally, high myopia, which was also present in our patients, can cause the reduction of ERG amplitudes [56,57,58].

A ring of increased autofluorescence is often seen in *RPGR*-cone/cone-rod dystrophy, delineating the border between affected and unaffected retina [27,30,32]. As has been observed previously [31,35,41], the rings in the studied patients enlarged with time, however, the ring enlargement was minimal (approx. 100 μm per year in diameter) suggesting a relatively stable disease after the initial loss of the foveal cones. Interestingly, the patient who had suffered from retinal detachment in the past appeared to have a smaller ring in the vitrectomized eye (Figure 1), however, the low quality of the FAF image on that eye could have had an influence on the measurement. In a recent study, Lima et al. reported three patients with cone-rod dystrophy, one of them harbouring *RPGR* mutation, and progressive expansion of the hyperautofluorescence ring area in 24-months follow-up [41]. The progression of the ring area in their study was faster (mean 9.2% per year) in comparison to ours (calculated at mean 2.9% per year). Their patients were younger (mean 21 years; range 18–25 years) and had smaller initial ring sizes (mean 1965 μm) in comparison to ours (mean 4231 μm). The difference in enlargement rate may reflect faster ring enlargement in early stages.

### 4.2. Female Carriers

Female carriers of the *RPGR* mutation are usually either asymptomatic or have delayed onset of the symptoms and show a milder phenotype [59]. Skewed inactivation of the X chromosome is thought to be responsible for manifesting more severe phenotype [44,60], however, other modifying genetic factors have also been suggested [61]. *RPGR* carriers appear to have four main patterns of fundus appearance: Normal or near normal pattern, a tapetal reflex, focal or patchy pigmentary retinopathy limited to a quadrant or hemisphere, and three or more quadrants of bone spicule pigmentation or atrophy [62]. Comander et al. reported 101 females from a family with *RPGR* mutation, showing that 40% of carriers had at least one abnormal test assessing visual acuity, visual field, or dark adaptation, with those carrying variants in *RPGR^ORF15^* having lower 30 Hz amplitudes on ERG compared to carriers with variants in exons 1–14 [62]. Recently, Talib et al. described the phenotypic spectrum of 125 female carriers of *RPGR* mutations from 49 pedigrees of RP and cone/cone-rod dystrophy (COD/CORD) [63]. The authors report a frequent (70%) occurrence of signs and/or symptoms in both, RP and COD/CORD carrier groups, with complete expression of RP or CORD in 29 heterozygotes (23%), carriers of *ORF15* mutations [63]. Both studies reported worse visual function in carriers of *RPGR^ORF15^* compared to carriers of the *RPGR* exon 1–14 mutations [62,63]. In our case, there was no family history of retinal disease in female carriers except for mild myopia in the mother of Patient 1. Nevertheless, the female carriers were not examined thoroughly at our institution, therefore a mild phenotype cannot be excluded. Considering the frequent occurrence of retinal disease in female carriers, it is important that they undergo regular ophthalmic follow-up because of the possible late-onset of the disease.

### 4.3. Haplotype Analysis

The results of the SNP analysis of probands from each family (Figure 2C) suggest that the *RPGR* variant is present on a common haplotype and may therefore originate from a common ancestor (founder effect).

### 4.4. RPGR C-Terminal Extension Variants’ Pathogenicity

*RPGR* variants resulting in the loss of the stop codon have not been conclusively established as a pathogenic mechanism in this gene. A single variant affecting the stop codon has been reported to date (c.3458A>C; Ter1153Serext*38 [30]), however, no convincing segregation or functional evidence has been provided to support its pathogenicity. The identification of two individuals with an overlapping phenotype and presence of the variant affecting the *RPGR* stop codon contributes novel evidence in support of the pathogenicity and further establishes this type of variants as pathogenic mechanism in this gene. According to the literature, it is still not clear whether all the loss of function *RPGR* variants result in retinal degeneration through the loss of protein function or gain a new one. Hong et al. suggested that some variants may cause a severe phenotype due to a dominant gain of function mechanism, resulting from the accumulation of truncated products [64]. Our stop loss variant extends the protein sequence with additional 38 residues, which may lead to a gain of a novel toxic or disruptive function. Therefore, our study also adds to the evidence for the pathogenic role of gain of function variants in *RPGR*.

### 4.5. Review of RPGR-Interacting Proteins Associated with Cone or Cone-Rod Dystrophy

The pathogenesis behind the *RPGR*-retinopathy is not clear [28,65]. The RPGR isoforms are localized in the connecting cilium of both cone and rod photoreceptors [28] and contain RCC1-like domains which facilitate interaction with other proteins [4,7,64,66]. Mutations within these domains may affect protein interactions [4] and with that transport between inner and outer segments [67]. The RPGR protein interacts with various ciliary proteins, such as the δ subunit of rod cyclic guanosine monophosphate phosphodiesterase (PDE δ), structural maintenance of chromosomes 1 and 3 (SMC1 and SMC3), GTPase Rab8A, whirlin, gelsolin, ARL3, INPP5E, RPGRIP1L (RPGRIP1 like), NPHP1, NPHP4, NPHP5 (IQCB1), NPHP6 (CEP290), and TTLL5 [12,16,68,69,70,71,72,73,74,75,76,77,78,79] in the mammalian retina. These protein interactions are known to differ among rods and cones, and certain mutations may have more effects on cone interactions than those of rods and vice versa [28]. Some insight into the pathogenesis behind *RPGR*-cone/cone-rod dystrophy may be gained by studying the proteins that interact with RPGR and also associate with the phenotype above. One of those is *RPGRIP1*, known to associate with LCA (Leber congenital amaurosis) [80,81,82] as well as childhood-onset cone-rod dystrophy [83]. Recently, cone/cone-rod dystrophy was also described in patients harbouring mutations in TTLL5 (tubulin tyrosine ligase like-5) and ARL3 (Arf-like protein 3), both interacting partners of RPGR [76,84]. TTLL5 localizes to the ciliary base and is important for the glutamylation of the ORF15 [85]. Glutamylation of the RPGR^ORF15^ as posttranslational modification is critical for its function in photoreceptors [76]. ARL3 (ADP-ribosylation factor-like protein 3), a small G protein, is mainly situated on the connecting cilium myoid region of the inner segments of cone photoreceptors and acts as an allosteric factor for the release of lipidated proteins bound to PDE6δ (δ unit of cGMP phosphodiesterase) [86]. Loss of ARL3 function may impair the trafficking of the lipidated outer segment proteins, leading to outer segment shortening and slow retinal degeneration [87]. Previously, missense variants in ARL3 were reported to cause Joubert syndrome, characterized by hypoplasia of the cerebellar vermis, developmental delay, renal anomalies, and rod-cone dystrophy [88]. However, in a recent study, a novel missense variant p.(Arg99Ile) in *ARL3* has been described, resulting in a cone-rod dystrophy [86]. *CEP290 (NPHP6)* is primarily involved in syndromic and non-syndromic LCA [89], however, a rare form of cone-dominated retinal dystrophy associated with mutations in *CEP290* has recently been described in two siblings [90]. They were firstly diagnosed with oligocone trichromacy (OT), which is thought to be a stationary condition, but then converted into a progressive degenerative disease [90]. It has been hypothesized that differing phenotypes indicate different functions of CEP290 in rods and cones [91] or cones being more vulnerable compared to rods due to their higher metabolism [92]. Similar reasons could be behind the phenotypic spectrum of *RPGR*-retinopathy. A novel insight into RPGR function has recently been reported [93]. By studying the cilia in patient’s fibroblasts, the researchers showed that *ORF15* mutation resulted in cilia length defects, mutant mRNA instability, and in some cases a significant increase in the RPGR 1–19/RPGR ORF15 ratio. The authors proposed that the relative levels of both RPGR isoforms are critical for optimal cilia growth as overexpression of the RPGR^ex1–19^ resulted in longer cilia, while that of RPGR^ORF15^ resulted in shorter cilia [93]. These observations could be of importance in the development of gene therapy, which is currently underway [93,94]. Further clinical and in vitro studies are needed to fully elucidate the complexity of RPGR function in the retina.

## 5. Conclusions

The present study describes the clinical phenotype of three patients harbouring a mutation in the terminal codon of *RPGR^ORF15^*. The phenotype of cone dystrophy in all three patients is consistent with previous observation that mutations in the distal end result in cone/cone-rod phenotypes. Furthermore, the fact that proximal mutations in *ORF15* exon cause RP, the terminal mutation Ter1153Lysext*38 cone-dystrophy and mutations in between cone or cone-rod dystrophy, we propose that the involvement of rods diminishes downstream of the *ORF15* exon.

## Figures and Tables

**Figure 3 genes-12-00499-f003:**
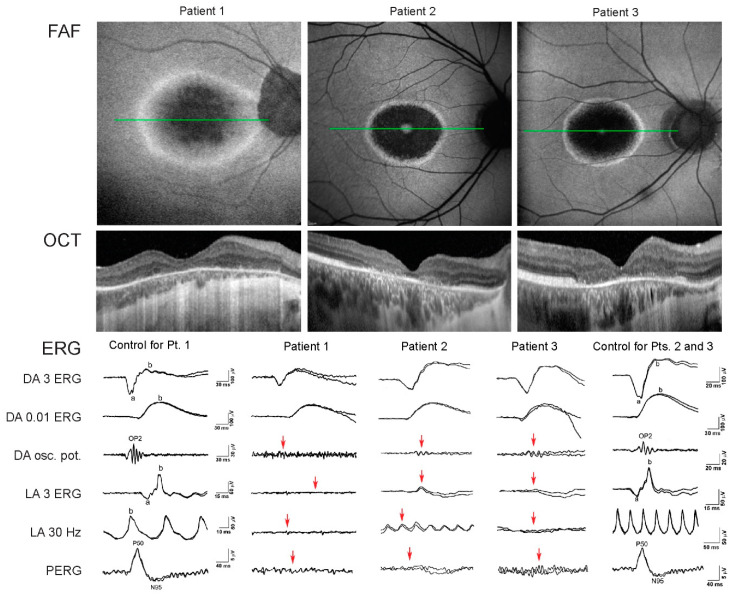
FAF, OCT, and ERG of the three studied patients. Note that although the brothers (Patients 2 and 3) had similar FAF and OCT phenotypes, Patient 2 had better LA ERG and residual PERG response coinciding with better visual acuity.

**Figure 4 genes-12-00499-f004:**
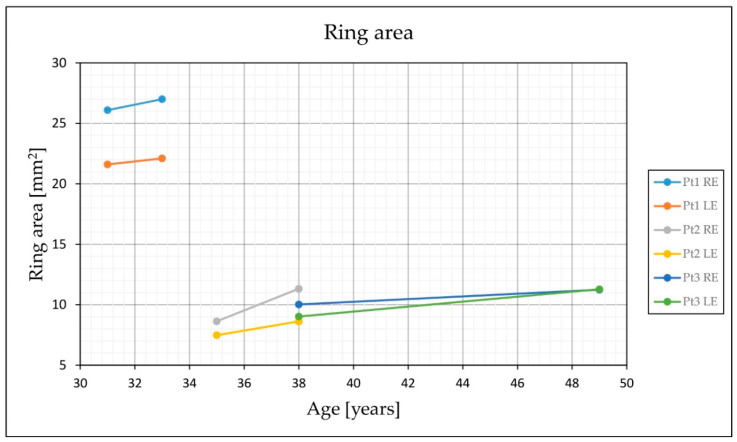
Enlargement of the hyperautofluorescent ring area with time. Enlargement of the hyperautofluorescent ring was observed in all patients (see also Figure 1). Ring area has been observed and measured between first and last examinations in all patients. Exact measurements are given in Table 1.

**Table 1 genes-12-00499-t001:** Clinical characteristics of the included patients. Data from the first and last exam are stated when they differed.

Patient ID	Age at the First and Last Examination (years)	Age at Onset	Ishihara	Refraction (Dioptre)		BCVA, logMAR (Snellen Decimal)		Visual Field	Fundus Features	Fundus Autofluorescence	OCT	Ring Area [mm^2^] (Ring Diameter [μm])		Electroretinography
			BE	RE	LE	RE	LE					RE	LE	
1	31; 33	Childhood: Refraction error (myopia), early 30s: loss of central vision, photophobia	1/15	−17.00–3.00/75°	−2.25–0.75/11° (pseudophakic eye after vitrectomy)	1.5 (0.03); 1.6 (0.02)	2.1 (0.0075); 1.5 (0.03)	Central scotoma	BE: Bull’s eye appearance of macula, optic pallor, attenuated vessels. LE: Bone spicules in the central and peripheral retina (after retinal detachment)	BE: Hyperautoflourescent ring; LE: RPE mottling in the inferior temporal retina (after retinal detachment)	BE: Absent RPE, Ise, and ELM in the central macula	26.1 (6342); 27.0 (6635)	21.6 (4267); 22.1 (4280)	BE: Undetectable PERG, normal DA ERG, undetectable LA ERG, significantly reduced mfERG
2 *	35; 38	Childhood: refraction error (myopia), early 30s: Loss of central vision, difficulties in colour discrimination and night blindness	1/15	−2.25–1.0/34°	−2.0–0.5/139°	0.3 (0.5); 0.7 (0.2)	0.2 (0.6); 0.5 (0.3)	Central scotoma	BE: Bull’s eye appearance of macula, optic pallor, attenuated vessels	BE: Hyperautofluorescent ring	BE: Absent RPE, Ise, and ELM in the central macula LE: Remnants of ELM in the foveola	8.6 (3618); 11.3 (4284)	7.5 (3462); 8.6 (3623)	BE: Significantly reduced PERG, normal DA ERG, significantly reduced and delayed LA ERG, reduced mfERG
3 *	38; 49	Childhood refraction error (myopia), early 30s: Photophobia, difficulties in colour discrimination	1/15	−12.0–2.0/80°	−14.0–4.0/90°	2.1 (0.0075); 1.5 (0.03)	1.8 (0.015); 1.5 (0.03)	Central scotoma	BE: Bull’s eye appearance of macula, optic pallor, attenuated vessels	BE: Hyperautofluorescent ring	BE: Absent RPE, Ise, and ELM in the central macula with remnants of the ELM in the foveola	10 (3917); 11.2 (4100)	9.0 (3778); 11.3 (4081)	BE: Undetectable PERG; normal DA ERG, significantly reduced to undetectable LA ERG

Abbreviation explanation: LE—left eye, RE—right eye, BE—both eyes, BCVA—best corrected visual acuity, CC—cum correctione, P CC—partial cum correctione, OCT—optical coherence tomography, ERG—electroretinography, mfERG—multifocal ERG, PERG—pattern ERG, LA—light adapted, DA—dark adapted. * Brothers.

## Data Availability

The data presented in this study are available on request from the corresponding author. The data are not publicly available due to personal data protection.

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
