# Peer review of "Cone Dystrophy Associated with a Novel Variant in the Terminal Codon of the RPGR-ORF15"

_genes, 2021, doi:10.3390/genes12040499_

Round 1

Reviewer 1 Report

The authors use exome sequencing to locate a potential mutation in a candidate gene that may cause the observed ophtalmological impairments of the three patients and come up with a mutation in the stop codon in the 15th exon of the gene RPGR, which leads to a nonsense mutation. Due to the nonsense mutation the stop codon is replaced by a lysine and the polypeptide is extended by 38 further aminoacids. Note that this mutation is a nonsense and not a frameshift mutation, as the authors claim. (A frameshift results from an insertion or deletion polymorphism of nucleotides that are not multiples of three.) The pedigrees in the two families are consistent with the phenotype, as the gene is located on the the X and the patients are males. At first, I thought that the molecular evidence the authors present is too thin, but a read-through of the terminal codon cannot really be studied on the RNA level and rather requires study of the resultant protein. Requesting such an analysis would be too difficult. I therefore agree with the authors that the molecular genetic part is adequate.

Unfortunately the English language requires many corrections, eg, articles are often not placed correctly. I corrected some mistakes in the manuscript (attached), but many will have slipped my notice.

Author Response

Dear Reviewer, thank you very much for your comments and suggestions. We will reply point-by-point to your questions listed.

Point 1: The authors use exome sequencing to locate a potential mutation in a candidate gene that may cause the observed ophtalmological impairments of the three patients and come up with a mutation in the stop codon in the 15th exon of the gene RPGR, which leads to a nonsense mutation. Due to the nonsense mutation the stop codon is replaced by a lysine and the polypeptide is extended by 38 further aminoacids. Note that this mutation is a nonsense and not a frameshift mutation, as the authors claim. (A frameshift results from an insertion or deletion polymorphism of nucleotides that are not multiples of three.) The pedigrees in the two families are consistent with the phenotype, as the gene is located on the the X and the patients are males. At first, I thought that the molecular evidence the authors present is too thin, but a read-through of the terminal codon cannot really be studied on the RNA level and rather requires study of the resultant protein. Requesting such an analysis would be too difficult. I therefore agree with the authors that the molecular genetic part is adequate.

Response 1: We must agree the mutation described is a nonsense mutation, actually it is a C-terminal extension variant due to the loss of the terminal RPGRORF15  codon. We must have been mistaken during writing – it has been corrected so far. Due to the misnamed mutation, we have changed the title of our article (please see lines 2 to 4 in the file attached), the new title is: Cone dystrophy associated with a novel C-terminal extension variant c.3457T>A (Ter1153Lysext*38) due to the loss of the terminal RPGRORF15 . According to the new title we have also changed some lines in the text (please see lines 21 – 24, 135 – 137, 191 – 195, 235 – 237 in the file attached).

Point 2: Unfortunately the English language requires many corrections, eg, articles are often not placed correctly. I corrected some mistakes in the manuscript (attached), but many will have slipped my notice.

Response 2: English language has been corrected appropriately (please see file attached). According to the grammar mistakes we did during writing, some sentences have been changed in our manuscript, so please see lines 25 – 27, 28, 73-75, 104, 110 – 116, 120, 126 – 130, 141 – 142, 152 – 154, 158 – 159, 190, 224 – 227, Table 1, Figure 4 (description below the graph), 264, 270, 276, 278, 290, 297, 302, 304, 311, 332, 336, 341 and 348 in the file attached.

Thank you very much for the consideration. 

Reviewer 2 Report

I find this case report very interesting and I think the paper is important in extending the knowledge on the field. From the genetic point of view the methodology is sastisfactory, terminology is correct due to HGVS recommendations, mutation assesment meets ACMG/AMP standards, description of the case is very good.

I have some minor remarks:

  1. I find following sentence in the abstract rather abundant: „Filtered variants were analyzed according to the population frequency, ClinVar database, the variant's putative impact, and predicted pathogenicity and were classified according to the ACMG guidelines.“ I would use the space in the favor of the results.
  2. I’m not sure about the meaning of this this sentence: „The variant is yet been reported (and is not present in the GnomAD...)“ does it mean that „the variant hasn’t been reported yet“ or „the variant has been reported but is not in the gnomAD yet“?
  3. There are 2 families from Slovenia described with the same mutation that hasn’t been described anywhere else before. Is there any sign that these families are related to each other? Has STR marker analysis been performed to confirm the founder effect?

Author Response

Dear Reviewer, thank you very much for your comments and suggestions. We will reply point-by-point to your questions listed.

Point 1: I find following sentence in the abstract rather abundant: „Filtered variants were analyzed according to the population frequency, ClinVar database, the variant's putative impact, and predicted pathogenicity and were classified according to the ACMG guidelines.“ I would use the space in the favor of the results.

Response 1: The sentence: ,,Filtered variants were analyzed according to the population frequency, ClinVar database, the variant's putative impact, and predicted pathogenicity and were classified according to the ACMG guidelines'' has been modified to: ,,Filtered variants were analyzed according to the population frequency, ClinVar database, the variant's putative impact and predicted pathogenicity; and were classified according to the ACMG guidelines'' (please see lines 25-27 in the file attached).

Point 2: I’m not sure about the meaning of this this sentence: „The variant is yet been reported (and is not present in the GnomAD...)“ does it mean that „the variant hasn’t been reported yet“ or „the variant has been reported but is not in the gnomAD yet“?

Response 2: We have modified a little the sentence that you asked about the meaning: „The variant is yet been reported (and is not present in the GnomAD...)“ to: ,,The variant has not been reported in the biomedical literature and is not present in the GnomAD population database'' (please see lines 194-196 in the file attached).

Point 3: There are 2 families from Slovenia described with the same mutation that hasn’t been described anywhere else before. Is there any sign that these families are related to each other? Has STR marker analysis been performed to confirm the founder effect?

Response 3: Single-nucleotide polymorphism (SNP) analysis comparing X-chromosomes of Patient 1 (Family 1) and Patient 2 (Family 2) has been performed in order to inspect the possibility of a founder effect. What we found out was that the X-chromosomes differed in the majority of the segments, except in one where they involved the same hemizygous variants. These findings suggest that the studied variant is being transmitted on the same haplotype and our patients could have inherited the variant from a shared ancestor (for details please see Figure 1 in the file attached). According to the SNP analysis performed, we added an image to the Figure 1 (Figure 1-C), showing the comparison of the X-chromosome variants of probands from each family, with a description below. We also added a sentence in lines 167 – 169, 196 – 202 and a new subsection paragraph in lines 293 – 296 (please see file attached).

Thank you very much for the consideration.
